METHODS

# Accurate detection of shared genetic architecture from GWAS summary statistics in the small-sample context

**Thomas W. Willis**[1]*, **Chris Wallace**[1,2,3]

**1** MRC Biostatistics Unit, University of Cambridge, Cambridge, United Kingdom, **2** Cambridge Institute of Therapeutic Immunology and Infectious Disease, University of Cambridge, Cambridge, United Kingdom, **3** Department of Medicine, University of Cambridge, Cambridge, United Kingdom

* thomas.willis@mrc-bsu.cam.ac.uk

## Abstract

Assessment of the genetic similarity between two phenotypes can provide insight into a common genetic aetiology and inform the use of pleiotropy-informed, cross-phenotype analytical methods to identify novel genetic associations. The genetic correlation is a well-known means of quantifying and testing for genetic similarity between traits, but its estimates are subject to comparatively large sampling error. This makes it unsuitable for use in a small-sample context. We discuss the use of a previously published nonparametric test of genetic similarity for application to GWAS summary statistics. We establish that the null distribution of the test statistic is modelled better by an extreme value distribution than a transformation of the standard exponential distribution. We show with simulation studies and real data from GWAS of 18 phenotypes from the UK Biobank that the test is to be preferred for use with small sample sizes, particularly when genetic effects are few and large, outperforming the genetic correlation and another nonparametric statistical test of independence. We find the test suitable for the detection of genetic similarity in the rare disease context.

## Author summary

The genome-wide association study (GWAS) is a method used to identify genetic variants which contribute to the risk of developing disease. These genetic variants are frequently shared between conditions, such that the study of the genetic basis of one disease can be informed by knowledge of another, similar disease. This approach can be productive where the disease in question is rare such that a GWAS has less power to associate variants with the disease, but there exist larger GWAS of similar diseases. Existing methods do not measure genetic similarity precisely when patients are few. Here we assess a previously published method of testing for genetic similarity between pairs of diseases using GWAS data, the 'GPS' test, against three other methods with the use of real and simulated data. We present a new computational procedure for carrying out the test and show that the GPS test is superior to its comparators in identifying genetic similarity when the sample size is small and when the genetic similarity signal is less strong. Use of the test will enable

**Data Availability Statement:** Statistics produced in the analysis of real and simulated data sets in this paper have been deposited at https://doi.org/10.5281/zenodo.7150454. UK Biobank GWAS summary statistics were downloaded from www.

nealelab.is/uk-biobank. Reference data from Phase 3 of the 1000 Genomes Project were obtained from the Project's FTP server at http://ftp.1000genomes.ebi.ac.uk. LD scores were downloaded from https://data.broadinstitute.org/alkesgroup/LDSCORE/eur_w_ld_chr.tar.bz2. A snakemake-based pipeline is provided to automate the download and generation of the data used or produced in this work at https://github.com/twillis209/gps_paper_pipeline.

**Funding:** TW is funded by the Medical Research Council (MRC) https://mrc.ukri.org/ (MC UU 00002/4). CW is funded by the Wellcome Trust https://wellcome.ac.uk/ (WT107881), the Medical Research Council (MRC) https://mrc.ukri.org/ (MC UU 00002/4) and supported by the NIHR Cambridge BRC https://cambridgebrc.nihr.ac.uk/ (BRC-1215-20014). The funders had no role in study design, data collection and analysis, decision to publish, or preparation of the manuscript. The views expressed are those of the author(s) and not necessarily those of the NHS, the NIHR or the Department of Health and Social Care.

**Competing interests:** I have read the journal's policy and the authors of this manuscript have the following competing interests: CW receives funding from GSK and MSD, and is a part-time employee of GSK. These funders had no involvement in this work.

accurate detection of genetic similarity and the study of rarer conditions using data from better-characterised diseases.

## Introduction

Genetic pleiotropy is the association of a genetic variant with more than one trait and is a pervasive property of the human genome [1]. A consequence of this phenomenon is the sharing of causal variants between phenotypes. The relationship of multiple phenotypic characters to a common heritable factor was first inferred from observations of the coinheritance of certain traits and diseases. Of these, particularly memorable are Darwin's remarks that hairless dogs have imperfect teeth [2] and that blue-eyed cats are 'invariably deaf' [3, 4].

Genome-wide association studies (GWAS) have revealed the extent of human genetic pleiotropy. A 2011 systematic review estimated that 4.6% of SNPs and 16.9% of genes had cross-phenotype effects [1]. These were likely underestimates given the incomplete catalogue of significantly associated SNPs surveyed, the review's stringent criteria, and the growth in the number of published GWAS in the decade since [5]. Pleiotropy is frequently identified among groups of diseases exhibiting similar pathobiology. Work by Cotsapas and colleagues found that 44% of immune-mediated disease risk SNPs across seven conditions were associated with multiple immune-mediated diseases [6]. Sharing of genetic architecture has also been observed among psychiatric illnesses, most prominently between bipolar disorder and schizophrenia [7, 8].

Cross-phenotype effects contribute to observed phenotypic correlations. Hazel introduced the notion of partitioning the phenotypic correlation of two characters into an additive genetic component, the coheritability, and a joint non-additive genetic and environmental component, the coenvironmentability [9, 10]. The genetic correlation is a heritability-standardised form of coheritability and quantifies the sharing of additive genetic effects between traits. First estimated in groups of related individuals in twin and family studies, in the genomic era genetic correlation can be quantified in large cohorts of unrelated individuals using individual-level genotype data or genetic effect estimates from GWAS [11]. More precisely, these methods typically estimate the 'SNP-correlation', however, not the genetic correlation: their estimates include only the effects of common SNPs and exclude those of rarer SNPs and other forms of genetic variation, such as indels and structural variants.

In 2010 Yang and colleagues introduced a linear mixed model (LMM) approach to SNP-heritability estimation which made use of all genotyped SNPs, not just those attaining genome-wide significance [12]. This method is alternately called the genome-based restricted maximum likelihood (GREML) method or 'GCTA' after the accompanying software package [13]. It was later extended to the bivariate domain for the joint analysis of pairs of traits and estimation of their genetic correlation [14]. A drawback of the use of GCTA is its requirement for individual-level genotype data. Cross-trait linkage disequilibrium score regression (LDSC) allows the fast estimation of each trait's heritability and the genetic covariance of trait pairs using GWAS summary statistics instead [15, 16]. LDSC correlation estimates for a large number of phenotype pairs are now publicly available online [17–19]. The more recently published SumHer program, part of the LDAK software package, offers similar functionality to LDSC [20]. Summary statistic-based methods like LDSC and SumHer have the advantage that they do not require access to individual-level genotype data, which can be difficult to obtain.

The relative merits of these methods in genetic correlation estimation has received less attention in the discourse than has their capacity to estimate heritability. In general, the

sampling error of such estimates is relatively large as they combine estimates of coheritability and the heritability of both traits [21]. Wray et al. give the rule of thumb that at least 5,000 samples are needed per trait in order to identify as statistically significant a correlation of 0.2 with GCTA [22]. Guidance on the use of the LDSC software states that 5,000 samples are needed to achieve better than 'very noisy' results (https://github.com/bulik/ldsc/wiki/FAQ). The empirical standard errors of LDSC estimates are approximately twice as large as those of GCTA [11] and LDSC thus requires much larger sample sizes to achieve estimates of the same precision [23].

The requirement for a large sample size for more precise correlation estimation is problematic when studying rare diseases, which yield only small numbers of cases. A recent GWAS of a broad phenotype of antibody-deficient primary immunodeficiency had only 733 cases [24] (primary immunodeficiency has a minimum UK prevalence of around 1 in 17,000 [25]), whilst the deeply-phenotyped UK Biobank (UKBB) cohort, consisting of around 500,000 middle-aged and elderly participants, featured only 415 and 391 cases for GWAS of self-reported systemic lupus erythematosus (SLE) and schizophrenia [18, 26], respectively, to take just two indicative diseases with prevalence approximately 1 in 1,000 [27] and 4.6 in 1,000 [28], respectively. It is typical of biobank cohorts like those of the UKBB and FinnGen [29], which are not subject to ascertainment, to have far fewer cases of such diseases than cohorts expressly assembled for the study of a particular disease; by way of comparison, Bentham et al. mustered a total of 7,219 cases for their GWAS of SLE [30] and the Psychiatric Genomic Consortium's primary GWAS of schizophrenia had 74,776 cases [31]. The range of case numbers in biobank GWAS, summary statistics for which are often available in a uniformly processed and convenient format, do make such collections suitable for the construction of a test set of real GWAS on which to evaluate the performance of tests of shared genetic architecture across varied sample sizes. In this work we leverage one such collection, the Neale lab's 2018 UKBB GWAS release, for precisely this purpose. Small sample sizes motivate the search for procedures with which to test for the presence of genetic sharing between traits in the small-sample size context. The need is particularly acute when making use of cross-phenotype, pleiotropy-informed methods to mitigate the problem of small sample sizes by conditioning on more highly-powered studies of related phenotypes [32–38]. Selection of these related traits must be driven by knowledge of their genetic similarity to the primary trait in the first place.

The nonparametric genome-wide pairwise-association signal sharing (GPS) test was devised by Li and colleagues to determine whether two traits are genetically related using GWAS p-values from marginal tests of the association of SNPs with each trait [39]. The test was originally applied to the detection of genetic similarity between a collection of paediatric autoimmune diseases. As a nonparametric test statistic, the GPS test statistic is not an estimator of a biologically meaningful parameter like the genetic correlation. Instead it is only evidence against a null hypothesis of bivariate independence.

The GPS test is a candidate method for the detection of genetic similarity in the small-sample context. Here, we first establish the distribution of the GPS test statistic in the null case of bivariate independence, showing that it is better fit by a generalised extreme value distribution than a transformation of the standard exponential as originally proposed by Li and colleagues. We examine the GPS test, used with each of these null distributions, applied to simulated GWAS and real GWAS from the UKBB, comparing it to tests of the genetic correlation as estimated by LDSC and SumHer. We show that the GPS test is more powerful than its comparators in the small-sample context and that it controls the type 1 error rate when applied to simulated genetically uncorrelated phenotype pairs. We also show that the GPS test in this setting is superior to Hoeffding's test, a canonical test of bivariate independence [40].

## Description of the method

The GPS test evaluates a null hypothesis of bivariate independence for two random variables $U$ and $V$. Each random variable models the data-generating process from which p-values for tests of association of SNPs with a phenotype are drawn. We assume that a GPS test statistic which is improbable under this null hypothesis is evidence against the null hypothesis of no sharing of genetic architecture between the two phenotypes.

The GPS test statistic is computed using data vectors $u$ and $v$ which comprise p-values from GWAS of each phenotype testing the same SNPs, indexed $i = 1, \ldots, n$, for association. The pairs $\{(u_i, v_i)|i = 1, \ldots, n\}$ thus comprise SNP-indexed samples from $U$ and $V$. The test statistic $D$ is defined as follows [39]:

$$D = \sup_{(u,v)} \sqrt{\frac{n}{\ln n}} \frac{|F_{U,V}(u,v) - F_U(u)F_V(v)|}{\sqrt{F_U(u)F_V(v) - F_U(u)^2 F_V(v)^2}}, \tag{1}$$

where $F_{U,V}$ is the empirical bivariate distribution function of $(U, V)$, and $F_U$ and $F_V$ are the empirical distribution functions of $U$ and $V$, respectively. All three empirical cumulative distribution functions (ecdfs) are estimated from the data $u$ and $v$.

It was originally proposed that the reciprocal of the squared GPS test statistic $D$ is approximately distributed as a standard exponential random variable under the null hypothesis:

$$\frac{1}{D^2} \sim \text{Exp}(1). \tag{2}$$

We improve the method by assuming instead that the test statistic $D$ follows a generalised extreme value distribution (GEVD) under the null hypothesis, i.e.

$$D \sim \text{GEV}(a, b, c), \tag{3}$$

where $a$, $b$, and $c$ are the location, scale, and shape parameters of a GEVD to be determined. We label these two formulations of the test as the GPS-Exp and GPS-GEV tests, respectively.

We estimate the GEVD parameters using synthetic realisations of the GPS test statistic under the null hypothesis which we generate through a permutation procedure. We randomly permute the indices of the SNPs in one data vector $v$ to create a reordered vector $v^*$ and then compute the GPS statistic on $u$ and $v^*$ to obtain a null realisation. We repeat this procedure to generate 3,000 such realisations and then use maximum likelihood estimation to obtain parameter estimates.

We compute the GPS test statistic using a linkage disequilibrium (LD)-pruned set of SNPs as did Li and colleagues [39]. LD pruning reduces the degree of statistical dependence within a set of SNPs by removing one of each pair of SNPs with a squared correlation coefficient $r^2$ exceeding a specified threshold. In this work we estimated pairwise $r^2$ using haplotype data from European individuals from the Phase 3 release of the 1000 Genomes Project (1kGP) [41] and carried out LD pruning using PLINK v2.00a2.3LM [42, 43] (S2 Text). For our whole-genome simulation studies we retained Li and colleagues' choice of threshold $r^2 = 0.2$, but on the evidence of single-chromosome simulations (Verification and comparison) we chose $r^2 = 0.8$ for our analysis of real data sets (Applications). The choice of $r^2$ balances the need to capture association signals through the inclusion of many SNPs in a data set against the additional variance introduced into ecdf estimates through the use of dependent, covarying samples.

We provide an implementation of both GPS tests in C++ and R on GitHub (https://github.com/twillis209/gps_cpp). We developed a performant permutation procedure through the use

of fast bivariate ecdf algorithms which make practical the use of the GPS-GEV test for a large number of trait pairs.

## Verification and comparison

### The null distribution of the GPS statistic is better fit by the generalised extreme value distribution

We compared the suitability of the transformed standard exponential and GEV distributions for use as the GPS test's null distribution using p-values from GWAS of 18 UKBB traits published by the Neale lab [18] (S2 Text). For each pair of traits, we generated null realisations of the GPS test statistic using the permutation procedure described above then examined the uniformity of the p-values obtained from these test statistics by the GPS-Exp and GPS-GEV tests. We found that GPS-Exp test p-values were not uniformly distributed, in contravention of the expectation under the null (Fig 1).

We found the GEV distribution to provide a much better fit as judged by uniformity of its p-values under the null hypothesis (Fig 1B). The GEVD fit null realisations of the GPS statistic well, diverging only in the right tail where the empirical distribution's tail was lighter than that of the fitted GEVD (S1 Fig). The consequence of this behaviour is that the GPS-GEV test will generate conservative p-values for non-null values of the test statistic. We observed that the GEVD provided a better fit to p-values from data simulated under the null of bivariate independence also.

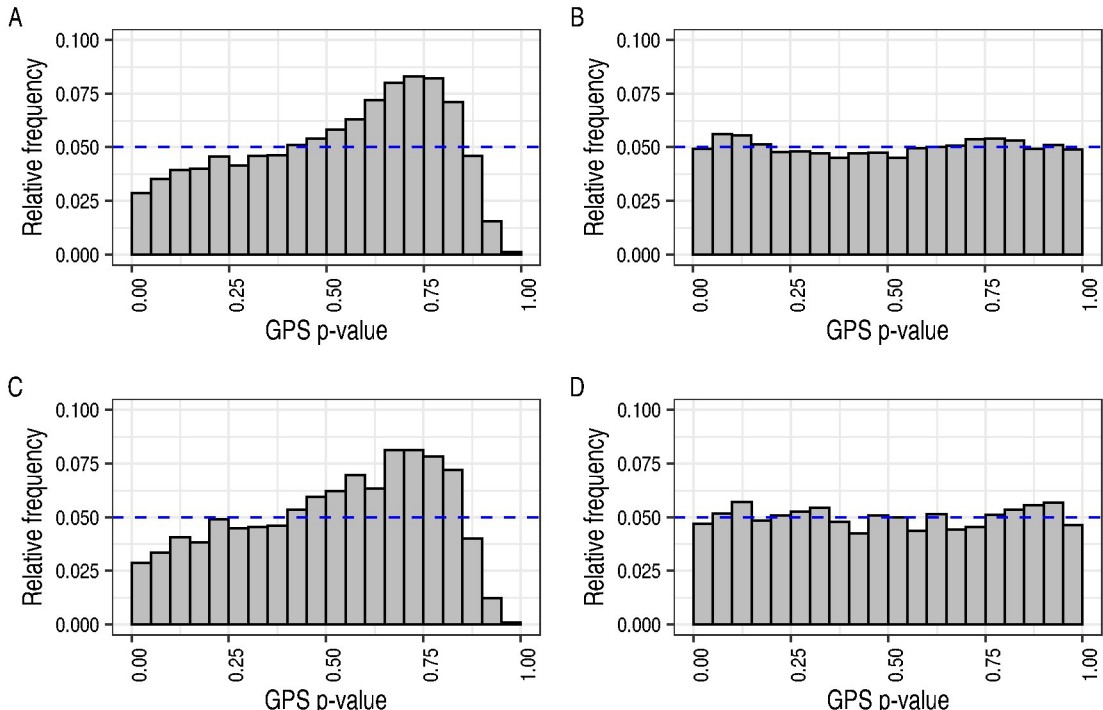

**Fig 1. GPS test p-values computed from the GEV distribution, but not thestandard exponential, were approximately uniform under the null.** (A) and (B) depict 10,000 GPS-Exp and GPS-GEV test p-values under the null, generated by permuting the order of SNPs for asthma in a pair of GWAS for asthma and emphysema/chronic bronchitis. (C) and (D) depict 10,000 GPS-Exp and GPS-GEV test p-values under the null, generated by permuting the order of SNPs in one of a pair of GWAS simulated under the small-effect architecture. The dashed line indicates the expected uniformity of p-values under the null.

Estimates of each parameter of the GEVD varied as a function of the number of SNPs (S2 Fig). We determined that, whilst the GEVD was the appropriate null distribution, it was necessary to obtain estimates of these parameters particular to each pair of data sets; in other words, it was not possible to determine parameter values for a single reference GEVD null distribution to which to compare GPS test statistic values for any pair of data sets.

Despite the use of fast ecdf algorithms the permutation procedure remained computationally burdensome. We thus sought to ascertain the smallest number of permutations needed to reliably estimate the GEVD parameters. We examined the value and precision of the estimates as a function of the number of permutations (S3 Fig). To determine how best to balance the need for accurate estimates against the cost of computation, we benchmarked three bivariate ecdf algorithms, the computation of the bivariate ecdf being by far the most costly step in calculating the GPS statistic. These were the naive algorithm making use of two nested loops, Langrené and Warin's recently published divide-and-conquer algorithm [44], and an order statistic tree-based algorithm from Perisic and Posse [45]. We found the last of these to be the fastest, offering a near-1000-fold advantage in speed over the naive algorithm and a 2-fold advantage over Langrené and Warin's algorithm in the bivariate case (Table A in S1 Text). The formulation of the permutation procedure is 'embarrassingly parallel' and we exploited this with a multithreaded implementation. Computing 3,000 GPS permutations for a 525,150-variant data set across 12 cores in parallel with Perisic and Posse's algorithm took approximately 18 minutes (or 3 hours and 17 minutes in CPU time). We chose 3,000 permutations as a compromise between estimate accuracy and running time. By way of comparison, the calculation of the GPS-Exp test p-value is computationally trivial, taking less than one second of CPU time.

## Simulation studies

To assess the ability of the two GPS tests to detect genetic similarity, we compared them to tests of non-zero genetic correlation using LDSC and SumHer (S2 Text). The heritability models we use for genetic correlation estimation with these tools make contrasting assumptions with respect to genetic architecture. The LDSC model considers only common SNPs with minor allele frequency (MAF) $> 0.05$ and assumes that per-SNP heritability is uniform across these SNPs; this uniform model, also termed the 'GCTA' model [46], is to be distinguished from the higher-dimensional LDSC models used in heritability estimation outside of the genetic correlation estimation context. By contrast, per-SNP heritability in SumHer's 'LDAK-Thin' model varies as a function of MAF and the level of linkage disequilibrium (LD) [20]. LDSC and SumHer do not use an LD-pruned SNP panel and instead define their input SNP panels in distinct ways (Table C in S1 and S2 Text).

We also compared the GPS tests to Hoeffding's test (S2 Text). Hoeffding's test is a nonparametric test of bivariate independence, like the GPS tests, and makes for an interesting comparator as a canonical, general-purpose test to set against the GPS tests developed specifically for the genomic context. Hoeffding's test assumes the use of independent observations, an assumption violated by our use of LD-pruned SNP sets with $r^2 > 0$ as discussed above. We nonetheless chose to apply Hoeffding's test to such LD-pruned sets to allow for a direct comparison with the GPS tests on the same input data.

We simulated GWAS data using `simGWAS v0.2.0-4` for synthetic case-control 'traits' with six different genetic architectures (Table 1) and five sample configurations across which the number of cases and controls, and case-control ratio varied (Table B in S1 Text). For each sample size and genetic architecture, we specified the number of causal variants to be shared between traits. For each configuration of architecture, case number, and number of shared

**Table 1. Details of the six whole-genome simulation regimes used to generate simulated GWAS summary statistics with `simGWAS`.**

| Name | Odds ratio(s) | No. of causal variants | No. of shared causal variants |
|---|---|---|---|
| Large-effect I | 1.2 | 25 | 5, 10, 15, 20, 25 |
| Large-effect II | 1.2 | 50 | 10, 20, 30, 40, 50 |
| Mixed-effect I | 1.05 and 1.2 | 200 (1.05) and 50 (1.2) | 100 (1.05) and 0, 15, 25 (1.2) |
| Mixed-effect II | 1.05 and 1.2 | 200 (1.05) and 50 (1.2) | 200 (1.05) and 0, 15, 25 (1.2) |
| Small-effect | 1.05 | 400 | 50, 100, 150, 200, 250, 300, 350, and 400 |
| Tiny-effect | 1.02 | 1,000 | 250, 500, 750, and 1,000 |

causal variants, we generated 200 replicate pairs of GWAS. For each replicate we computed the test statistic of each test under examination. Each simulated GWAS data set comprised 8,995,296 variants, although the number of variants taken as input depended upon the test in question (Table C in S1 Text).

As the induced heritability and coheritability depend not only on the effect size of the causal variant but also the minor allele frequency, which varied between the randomly chosen causal variants, it was not possible to strictly specify the genetic correlation across replicates, only the number of shared causal variants. As a result, replicates have close but non-identical values of the genetic correlation.

At $r^2 < 0.2$, all five tests controlled the type 1 error rate at or below the specified test size $\alpha = 0.05$ (Fig 2). Both SumHer and LDSC failed to produce valid genetic correlation estimates for a proportion of simulated data sets with small case numbers (S4 Fig): where SNP effect estimates are imprecise, SumHer and LDSC can return small, imprecise, and sometimes negative

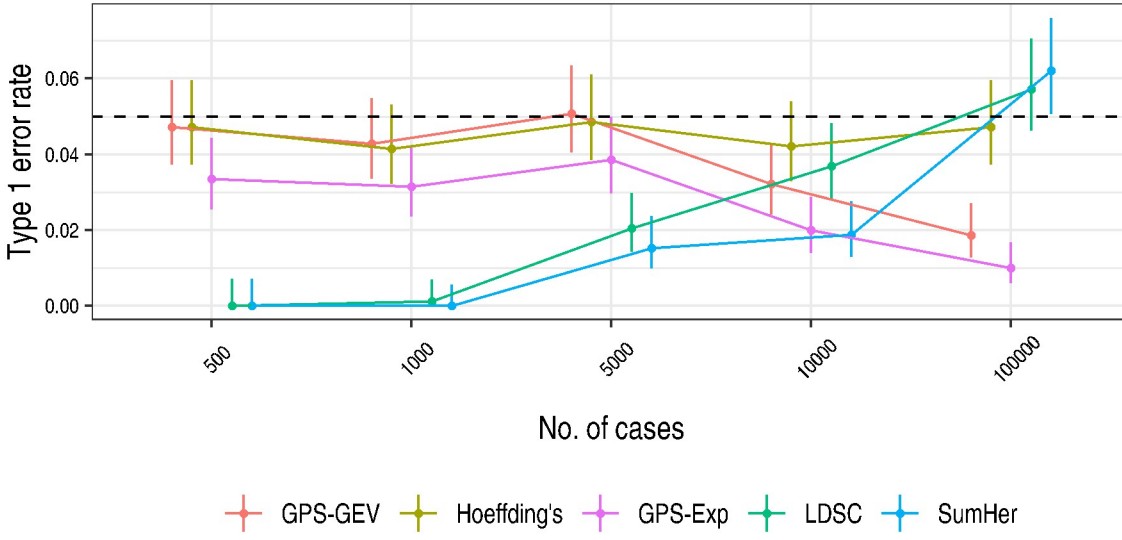

**Fig 2. The GPS tests provided for conservative control of the type 1 error.** The type 1 error rate for each method was estimated as the proportion of replicates for which $p \leq 0.05$ when the genetic correlation was set to zero. For LDSC and SumHer, this proportion was measured only among those replicates producing a valid test statistic; both methods may fail to return an estimate of the genetic correlation when variant effect estimates have large standard errors. 95% confidence intervals were calculated as Wilson score intervals. The dotted line depicts the size of the test, 0.05. Data was aggregated across simulation architectures at each sample size. The points at each number of cases have been spread along the x-axis by adding 'jitter' for the sake of clarity. A large proportion of replicates at case numbers smaller than 10,000 did not return valid test statistics for LDSC and SumHer (S4 Fig) which caused the deflated type 1 error rates observed here at lower case numbers.

heritability estimates which preclude estimation of the genetic correlation. For the three samples with the largest number of cases (5,000, 10,000, and 100,000), there was a downward trend in type 1 error for both GPS tests and we investigated this in detail (S3 Text).

Both GPS tests exhibited appreciable power to detect genetic similarity in simulated data sets with as few as 1,000 cases across four of the six simulation regimes (Fig 3). In the small-effect regime, their power remained low until the number of cases was increased to 10,000, at which size both GPS tests were outperformed by LDSC and SumHer. The GPS tests' power increased with the number of cases and the genetic correlation. The GPS-GEV test showed a

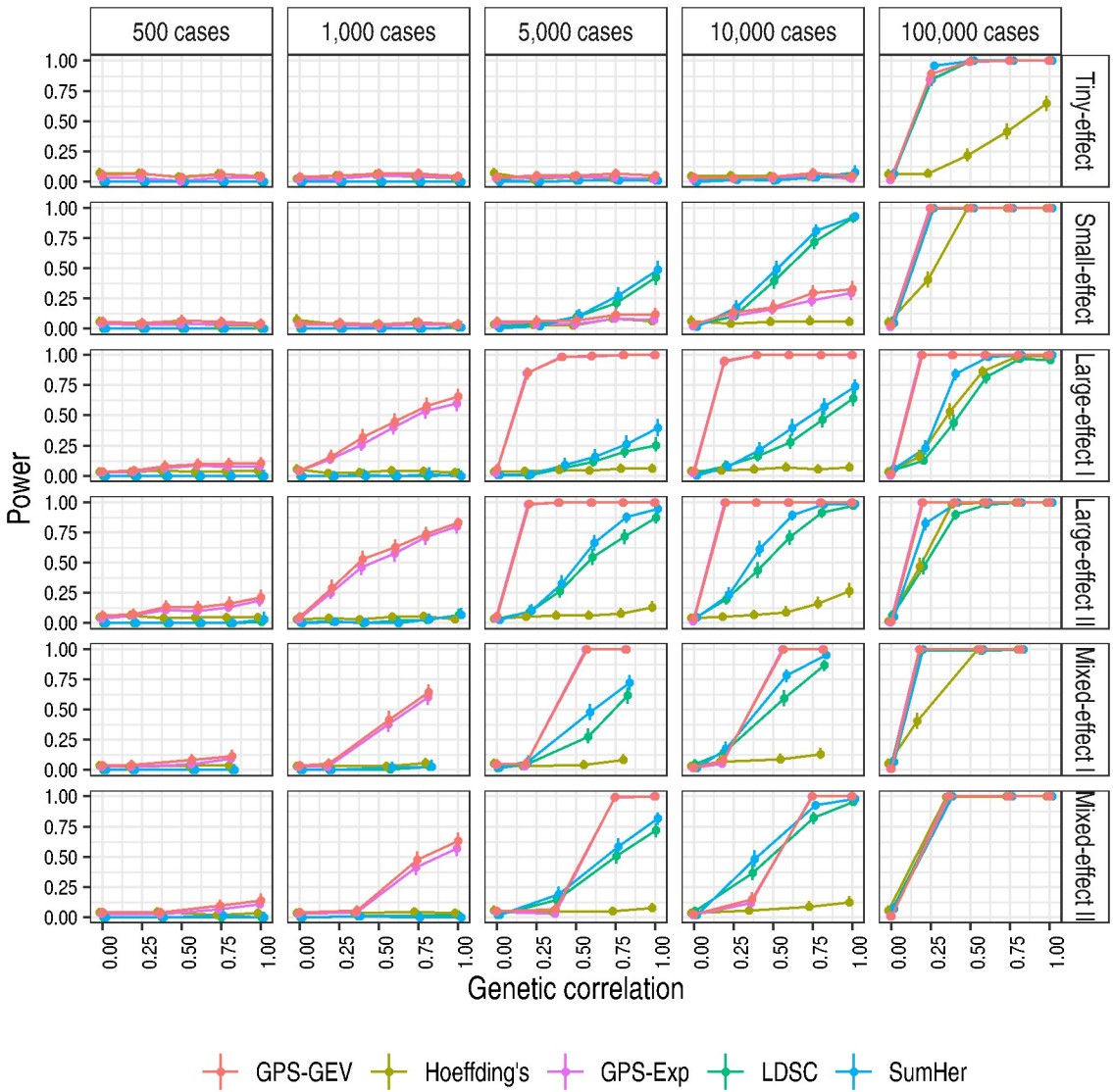

**Fig 3. The GPS tests showed an advantage in power over their comparators at smaller numbers of cases.** Power was estimated as the proportion of replicate pairs for which the p-value reached nominal significance, $p \leq 0.05$, for a specified number of shared causal variants and sample size. This proportion was measured only among those replicates producing a valid test statistic. 95% confidence intervals were calculated as Wilson score intervals. As genetic correlation varies across replicate pairs according to the precise shared causal variants randomly chosen, we grouped observations according to the number of shared causal variants. The value of the genetic correlation for a point is the mean of the specified genetic correlation across replicate pairs grouped for that point's estimate. To separate the points to depict the data more clearly, we added 'jitter' when drawing the figure.

marginal but consistent advantage in power over the GPS-Exp test, most notably at the 1,000-case sample size. We note also that LDSC and SumHer performed to a reassuringly similar degree despite the contrasting assumptions of their models of heritability.

As noted above, Hoeffding's test and both GPS tests are performed on an LD-pruned set of SNPs and the results presented thus far followed the original recommendation of Li and colleagues to prune using the threshold $r^2 = 0.2$. Pruning with this value maintains only a limited degree of dependence between SNPs in the pruned set, but may discard SNPs contributing the strongest signals in the data. The GPS statistic is computed as the supremum norm of a weighted distance between two distributions, rather than as an estimate of the integral of the squared distance as with Hoeffding's test statistic, or as a heritability-standardised estimate of genetic effect covariance across the genome as with the genetic correlation. The GPS tests might thus be particularly sensitive to the presence or absence of individual SNPs contributing strong signals, the LD pruning procedure selecting SNPs in an unbiased manner without reference to the significance of their test statistic. We explored the influence of $r^2$ on the performance of the tests which used LD-pruned sets.

We conducted additional simulation studies testing for multiple values of $r^2$ using two simulation regimes (Table D in S1 Text). We used the same range of sample configurations as before (Table B in S1 Text). For $r^2 = 0.5$ and 0.8, Hoeffding's test failed to control the type 1 error rate (S5 Fig). Hoeffding's test assumes independent samples and this assumption was violated by our use of $r^2 > 0$. For the GPS-Exp and GPS-GEV tests, type 1 error control was maintained, albeit with the same conservative trend with increasing sample size seen in the whole-genome simulations (S3 Text). We had no theoretical expectation of rising type 1 error as was the case with Hoeffding's test, but this was nonetheless surprising given the high degree of dependence expected at $r^2 = 0.8$. There was a large increase in power for all three tests as $r^2$ increased under the large-effect regime (Fig 4). Under the small-effect regime, power was considerably lower when compared with the large-effect regime but the same trend of increasing power for larger values of $r^2$ was observed. On the basis of these results, we chose $r^2 = 0.8$ for our studies on real data sets.

## Applications

The existence of pleiotropic loci manifesting cross-phenotype effects across many immune-mediated diseases is well-attested [6, 39, 47–50]. We surveyed UKBB disease traits to identify groups of immune- and non-immune-related diseases, including phenotypes with small, medium, and large numbers of cases (Table 2). Each phenotype was represented by a GWAS conducted on the same 13,791,467-variant panel by the Neale lab. We hypothesised that a sensitive test would be more likely to give a positive signal where both diseases belonged to the immune group than where diseases from the immune- and non-immune groups were paired together, or where both traits came from the group of largely unrelated non-immune diseases (collectively, the 'mixed' pairs).

Evaluating tests on real data sets precludes the specification of ground-truth values of the genetic correlation to assess power and type 1 error control. We instead assembled an additional collection of 18 GWAS, one for each of our chosen phenotypes, to provide out-of-sample estimates of the genetic correlation. We sought the largest publicly available GWAS to provide the most precise estimates possible (Table E in S1 Text). We term these the 'exemplary' data sets. The genetic correlation estimates thus obtained provided equivocal support for a greater sharing of genetic architecture among immune/immune pairs (S6 Fig). The median genetic correlation was higher for immune/immune pairs with a greater number of cases (0.15

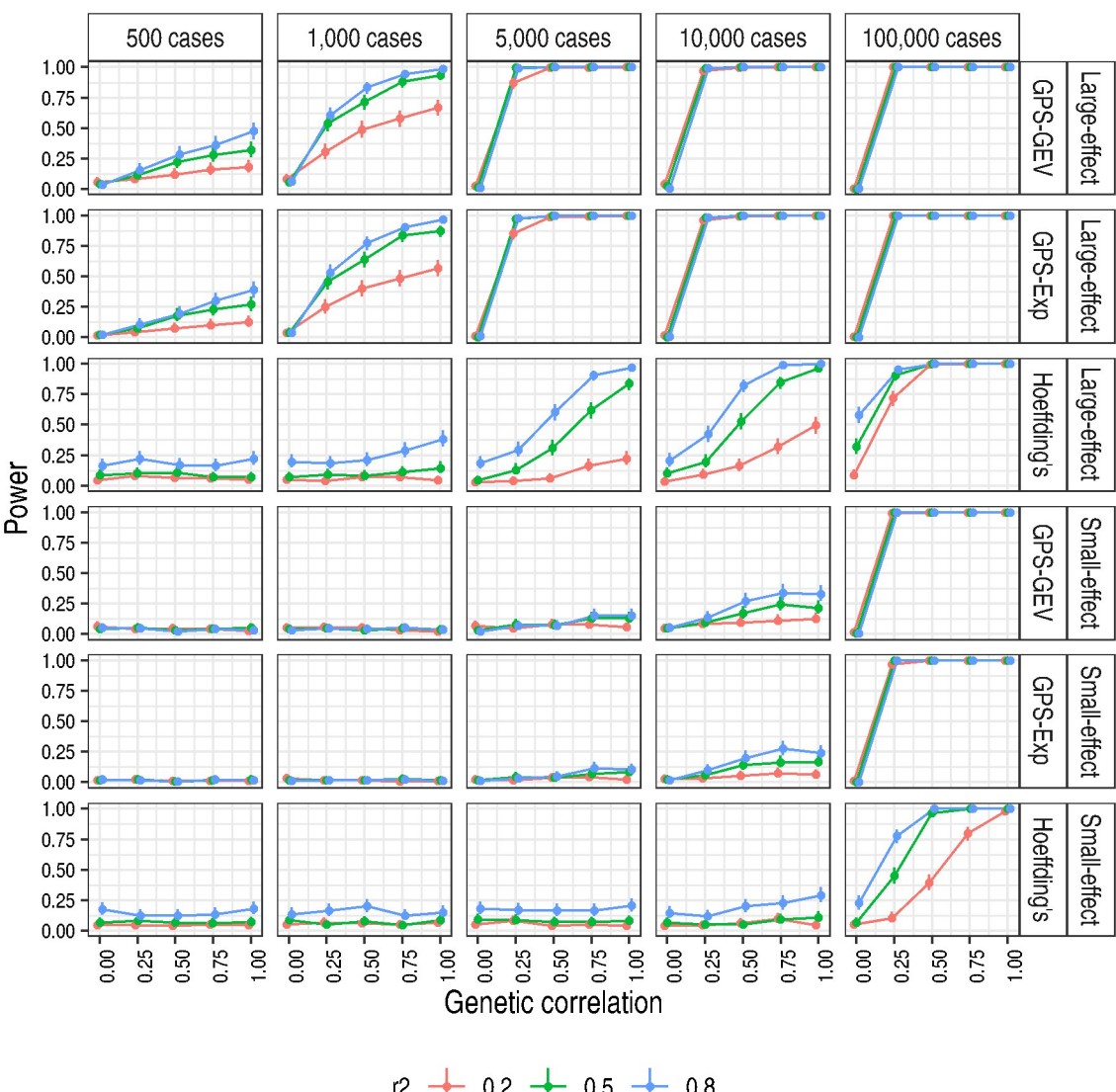

**Fig 4. Higher values of $r^2$ yielded an appreciable increase in power.** Power was estimated as the proportion of replicate pairs for which the p-value reached nominal significance, $p \leq 0.05$. 95% confidence intervals were calculated as Wilson score intervals. 'Small-effect' and 'Large-effect' refer to the simulation regimes in Table D in S1 Text. As genetic correlation varies across replicate pairs according to the precise shared causal variants randomly chosen, we grouped observations according to the number of shared causal variants. The value of the genetic correlation for a point is the mean of the specified genetic correlation across replicate pairs grouped for that point's estimate. To separate the points to depict the data more clearly, we added 'jitter' to the x-axis when drawing the figure.

compared with 0.07 for mixed pairs), but smaller for pairs with fewer cases (0.07 compared with 0.11 for mixed pairs) (Table F in S1 Text).

In light of these results we restricted our first analysis only to those immune/immune pairs with an exemplary genetic correlation estimate significantly different from 0 (S7 Fig), examining the proportion correctly identified as associated by each test. This group comprised 15 pairs with a median genetic correlation estimate of 0.27. We intended this as an approximate test of power in real data sets for which shared genetic architecture was biologically highly plausible. We performed tests with and without the inclusion of the major histocompatibility complex (MHC), a gene-rich locus which frequently exhibits highly significant association

**Table 2. The selected UK Biobank traits used to evaluate the performance of the GPS tests and their comparators.** The code of each trait specifies the precise phenotype and definition used from the UK Biobank collection.

| Trait | Abbreviation | Category | No. of cases | No. of controls | Code |
|---|---|---|---|---|---|
| lupus | - | immune | 415 | 360,726 | 20002_1381 |
| type 1 diabetes | T1D | immune | 583 | 360,611 | E4_DM1 |
| Crohn's disease | CD | immune | 1,096 | 360,045 | 20002_1462 |
| ulcerative colitis | UC | immune | 2,143 | 359,051 | K51 |
| rheumatoid arthritis | - | immune | 4,017 | 357,124 | 20002_1464 |
| eczema/dermatitis | - | immune | 9,321 | 351,820 | 20002_1452 |
| hypothyroidism | - | immune | 17,574 | 343,567 | 20002_1226 |
| hayfever | - | immune | 20,904 | 70,883 | 22126 |
| asthma | - | immune | 41,934 | 319,207 | 20002_1111 |
| cardiomyopathy | - | non-immune | 479 | 360,715 | I42 |
| endometriosis | - | non-immune | 1,496 | 359,698 | N80 |
| macular degeneration | MD | non-immune | 2,726 | 115,164 | 6148_5 |
| glaucoma | - | non-immune | 5,092 | 112,798 | 6148_2 |
| leiomyoma | - | non-immune | 5,507 | 355,687 | D25 |
| irritable bowel syndrome | IBS | non-immune | 8,537 | 352,604 | 20002_1154 |
| cholelithiasis | - | non-immune | 10,520 | 350,674 | K80 |
| osteoarthritis | - | non-immune | 30,046 | 331,095 | 20002_1465 |
| hypercholesterolaemia | - | non-immune | 43,957 | 317,184 | 20002_1473 |

signals in GWAS of immune diseases [51]. As noted above, the GPS statistic is constructed as the supremum norm of a weighted distance between distributions. This could plausibly render it more sensitive to the isolated coincidence of strong association signals for two traits at a single locus like the MHC in the absence of other evidence of genome-wide pleiotropy, as was seen in the early, relatively underpowered GWAS of immune diseases.

Informed by our simulation studies, we chose $r^2 = 0.8$ to construct LD-pruned data sets. We found the GPS tests had an advantage in power over their comparators, albeit on a small test set of 15 pairs which limited our capacity to draw confident inferences about test performance (Table 3 and S8 Fig). Inclusion of the MHC led to more powerful tests as expected.

We followed this power analysis with a study of type 1 error control by each test. As surrogate true negatives we used 62 mixed pairs with a non-significant exemplary genetic

**Table 3. The GPS tests performed best in immune/immune pairs but identified as associated a high proportion of ostensibly unrelated disease pairs.** The percentage of pairs for which a nominally significant ($p \leq 0.05$) test statistic was computed given by number of cases and MHC inclusion status, and the median genetic correlation estimate $\hat{r}_g$ in each group. 'No. of cases' gives the smaller number of disease cases in each pair of case-control UKBB GWAS. 'Median $\hat{r}_g$' denotes the median genetic correlation estimate obtained from the exemplary data sets for each group of pairs.

| Group | MHC | No. of cases | Median $\hat{r}_g$ | No. of pairs | GPS-GEV | GPS-Exp | Hoeffding's | LDSC | SumHer |
|---|---|---|---|---|---|---|---|---|---|
| immune | ✓ | $\leq 2,000$ | 0.15 | 7 | 100 | 100 | 86 | 29 | 43 |
| immune | ✓ | $> 2,000$ | 0.30 | 8 | 100 | 100 | 100 | 50 | 62 |
| immune | ✗ | $\leq 2,000$ | 0.15 | 7 | 71 | 71 | 43 | 29 | 29 |
| immune | ✗ | $> 2,000$ | 0.30 | 8 | 100 | 75 | 75 | 50 | 63 |
| mixed | ✓ | $\leq 2,000$ | 0.02 | 30 | 30 | 30 | 37 | 13 | 17 |
| mixed | ✓ | $> 2,000$ | 0.01 | 32 | 34 | 28 | 50 | 6 | 6 |
| mixed | ✗ | $\leq 2,000$ | 0.02 | 30 | 3 | 3 | 27 | 10 | 3 |
| mixed | ✗ | $> 2,000$ | 0.01 | 32 | 9 | 0 | 38 | 3 | 6 |

correlation estimate; the median exemplary genetic correlation estimate of this group was 0.01. Consistent with our simulation studies where choosing $r^2 = 0.8$ and violating Hoeffding's test's assumption of independent samples led to the loss of type 1 error control, Hoeffding's test identified a large number of ostensibly spurious trait associations among the mixed pairs (Table 3). Both GPS tests identified as associated approximately one third of pairs when the MHC was included, but its removal dramatically reduced the significant call rate to 0.06 and 0.02 for the GPS-GEV and GPS-Exp tests, respectively, when measured across jointly across both case number categories. Both LDSC and SumHer had significant call rates of approximately 0.1. Removal of the MHC rendered the GPS tests' significant call rate essentially comparable to that of LDSC and SumHer.

A possible explanation for the lower significant call rate of LDSC and SumHer was our use of the same parameter these methods estimate, the genetic correlation, to define our true negative set in the first place; indeed, we used SumHer's estimates to define this set, albeit those obtained from out-of-sample data sets. The GPS tests' higher significant call rate on these data sets was not consistent with our observation of their conservative performance on simulated data sets. The dramatic reduction in the significant call rate for both GPS tests effected by removal of the MHC was consistent with the dependence of these tests' significance on limited but strong sharing of association signals. Our ostensible 'true negatives' may well have exhibited shared genetic architecture within the MHC but not to an extent, within or without the MHC, sufficient to induce a nominally significant genetic correlation.

An alternative explanation may be dependence between data sets with very high fractions of shared controls and a smaller fraction of shared cases. This is typical of collections of GWAS performed using samples from the same biobank. Sharing of cases is especially likely to occur for GWAS of correlated phenotypes, such as the atopic diseases: asthma, eczema, and hayfever. We can estimate the expected correlation between effect estimates at null SNPs induced by this sample overlap per Lin and Sullivan [52] as

$$\rho = n_0 \sqrt{\frac{n_{k1}n_{l1}}{n_{k0}n_{l0}}} + n_1 \sqrt{\frac{n_{k0}n_{l0}}{n_{k1}n_{l1}}} \Big/ \sqrt{(n_{0k} + n_{1k})(n_{0l} + n_{1l})}, \tag{4}$$

where $n_0$, $n_1$ are the total numbers of shared controls and cases between two studies $k$ and $l$, $n_{0k}$ and $n_{1k}$ the number of controls and cases in study $k$, and $n_{0l}$ and $n_{1l}$ the number of controls and cases in study $l$. We estimated $\rho$ for all pairs of the UKBB data sets used, and found it ranged from 0.003 to 0.228, median 0.034 (S1 Data). While the correlation was higher for traits known to be comorbid (e.g. 0.086 for UC-CD), the largest correlations were estimated for pairs of traits with the largest case numbers; the correlation was 0.228 for osteoarthritis-hypercholesterolaemia, both of which have over 40,000 cases in the UKBB (S9 Fig). While we do not anticipate such case numbers in our envisioned use case for the GPS test, we conducted a simulation study to validate the UKBB results. We simulated pairs of Z scores for pairs of data sets where each data set held a mixture of truly associated and non-associated SNPs, but where none of the associated SNPs were shared, i.e. the case under the GPS test's null. We found that there was some inflation in the GPS's type 1 error rate when $\rho > 0.1$ and the power in both data sets was very low, i.e. where fewer than one genome-wide significant effect was expected in either data set (S10 Fig). As power increased however, we observed the same falling in type 1 error rate seen with increasing sample size above. This was sufficient to ensure expected type 1 error rates were controlled in all simulation scenarios where at least one genome-wide significant effect was expected, a threshold comfortably exceeded for all traits chosen for our UKBB analysis.

## Discussion

We conducted a comprehensive study of both GPS tests and found them to be a superior means of identifying genetic similarity between disease traits in the small-sample context. For simulated data, the GPS tests' power was greater than its comparators when the number of cases or genetic correlation was small. This greater power was also in evidence when applying the GPS tests to immune disease pairs from the UK Biobank. In simulations we found the GPS-GEV test to offer an advantage over the GPS-Exp test in terms of power at the cost of the computation required by its the permutation procedure. We also found use of a SNP panel pruned with a far higher value of $r^2$ to improve the power of the GPS tests without loss of type 1 error control.

With the exception of immune pairs with strong prior evidence of association, a significant GPS-GEV test statistic most often depended upon the inclusion of the MHC. Where the success of pleiotropy-informed methods depends upon the presence of broad genome-wide sharing of genetic effects across multiple loci, the test may be less useful in identifying suitable traits to leverage. Additionally, the presence of extensive LD in the MHC and the relatively high value of $r^2 = 0.8$ which we found to be most powerful mean that LD may allow distinct causal variants to drive spurious test results. A significant test statistic may arise from the presence of a SNP at which highly significant p-values coincide in each data set, but which is in fact in LD with causal variants distinct for each trait. This problem is compounded by the strength of MHC signals: p-values smaller than 1e-100 are frequently observed there. On the other hand, the discovery of shared MHC signals in early GWAS of immune-mediated diseases did precede later evidence of numerous shared causal variants in larger studies. Although this phenomenon may be particular to the genetic architecture of immune traits, the MHC acting as a sentinel of wider genetic similarity would go some way to vindicate this behaviour of the GPS tests, at least in their application to immune-mediated diseases.

The GPS test statistic's construction as a supremum norm seemed to explain its better performance in the large- and mixed-effect regimes compared with the small-effect regime and its sensitivity to MHC signals. The statistic is best maximised by the presence of shared large-effect variants, such as those seen at the MHC, than a cumulative signal comprised of many smaller-effect variants.

We speculate that pervasive genetic similarity might be better detected with the use of a test of bivariate independence based on a quadratic ecdf statistic, where the distance criterion is the integral of the squared distance between two cdfs. Hoeffding's test uses one such statistic, but its adaptation to incorporate the Anderson-Darling distance's weight function [53] could improve its use in the GWAS context. This weight function emphasises divergence in the tails of the cdfs, that is, it upweights the contribution of pairwise-dependent p-values where these are small and thus more significant. The GPS test statistic is a supremum norm of a weighted distance: the expression in its denominator is the square root of a bivariate function analogous to the univariate Anderson-Darling weight function and this component appears to accomplish just such an upweighting of distances observed at more significant p-values.

As nonparametric tests of bivariate independence, the GPS tests do not provide an estimate of an interpretable parameter like the genetic correlation. Whilst their null hypothesis can be precisely stated in statistical terms, the biological inference to be drawn from rejection of the null is unclear beyond a notion of some sharing of genetic effects between traits. As their test statistic is computed from p-values, the GPS tests are agnostic as to the direction of genetic effects: a significant test statistic can arise where a negative genetic correlation is observed between two traits, something we observed for hypercholesterolaemia and hayfever. Thus, whilst GPS tests may be more sensitive in small-sample cases, examination

of the sign of effect estimates at SNPs with small p-values in both traits can also be informative.

We recommend the GPS-GEV test for the detection of shared genetic effects with small-sample data sets where heritability-based methods are not suitable. We note that, despite our exploration of the effect of sample overlap on performance, we do not anticipate this will be common in its use case, which is likely to involve at least one bespoke GWAS of a rare disease. When samples are large enough to allow heritability-based approaches to be used, we would advocate them, particularly where they attempt to account explicitly for sample overlap as do LDSC and SumHer. It may be employed to detect genetic relationships between rare diseases and more common, better-understood traits or as a tool for the selection of genetically similar traits for analysis with pleiotropy-informed, cross-phenotype methods, which can be employed to overcome the problem of low power due to small sample size in association studies.

## Supporting information

**S1 Fig. The generalised extreme value distribution (GEVD) provided a conservative fit to realisations of the GPS test statistic under the null hypothesis.** (A) a histogram of GPS statistics with fitted GEVD density superimposed as the red line and (B) a quantile-quantile plot of empirical ('Sample') and fitted GEVD ('Theoretical') quantiles. Null realisations of the GPS statistic were produced by permutation of the order of SNPs in GWAS of asthma and emphysema/chronic bronchitis from the UK Biobank collection.
(TIF)

**S2 Fig. The dependence of estimates of the three generalised extreme value distribution parameters on the number of SNPs.** (A) The location parameter. (B) The scale parameter. (C) The shape parameter. The error bars depict 95% confidence intervals for each estimate. The estimates were obtained by permuting the order of SNPs in a pair of GWAS summary statistics data sets and downsampling these same data sets to obtain the specified number of SNPs. The estimates shown were obtained using GWAS of asthma and emphysema/chronic bronchitis in the UKBB data set.
(TIF)

**S3 Fig. The dependence of estimates of the three generalised extreme value distribution parameters on the number of permutations $n$ used to fit them.** (A) The location parameter. (B) The scale parameter. (C) The shape parameter. Error bars depict 95% confidence intervals for each estimate. The dashed blue line indicates 3,000 permutations. The estimates were obtained by permuting the order of SNPs in a pair of GWAS summary statistic data sets. The estimates shown were obtained using GWAS of asthma and emphysema/chronic bronchitis.
(TIF)

**S4 Fig. The proportion of simulated data sets for which a test statistic could not be obtained.** Failures were recorded only for LDSC and SumHer.
(TIF)

**S5 Fig. Type 1 error rate was not controlled for higher values of $r^2$ by Hoeffding's test.** The type 1 error rate for each method was estimated as the proportion of replicates for which $p \leq 0.05$ when the genetic correlation was set to zero. This proportion was measured among 200 replicates. 95% confidence intervals were calculated as Wilson score intervals. The dotted line depicts the size of the test, 0.05.
(TIF)

**S6 Fig. Genetic correlation estimates obtained from the exemplary data sets.** 'Corr' is estimated genetic correlation.
(TIF)

**S7 Fig. Genetic correlation estimates obtained from the exemplary data sets only for those pairs with nominally significant estimates.** A nominally significant estimate was taken as one for which $p \leq 0.05$ from a chi-squared test of non-zero genetic correlation. 'Corr' is estimated genetic correlation.
(TIF)

**S8 Fig. The GPS tests showed more power than their comparators to detect association between pairs of immune diseases with strong prior evidence of association.** The 'No. of cases' gives in terms of point size the smaller number of disease cases in each pair of case-control GWAS. The dashed line lies at 0.05. Filled points indicate the use of the entire data set, hollow points indicate the use of a data set with the MHC removed. Where p-values were smaller than 1e-8, they have been aliased to 1e-8 and indicated with an asterisk. For very small case numbers, it was not possible to obtain genetic correlation estimates with LDSC and SumHer for some data sets.
(TIF)

**S9 Fig. The estimated correlation of effect estimates $\rho$ was largest among those pairs with the largest number of cases.** The shade of the heatmap gives the value of $\rho$ for each pair of traits. The sample size of each GWAS is depicted in terms of its number of cases on the supplementary colour scale.
(TIF)

**S10 Fig. The type 1 error rate of the GPS tests was elevated only for larger values of $\rho$.** (A) GWAS p-values were simulated from a transformed mixture of normal distributions with null and non-null components. The distribution of null and non-null p-values is depicted in black and blue, respectively. 'zmean' and 'zsd' give the mean and standard deviation of the normal distribution from which non-null Z-score addends were drawn for each data set. The percentage of SNPs expected to exceed the genome-wide significance threshold of $5 \times 10^{-8}$ for each simulation is stated; this point is depicted by the vertical dashed line. (B) The type 1 error rate for each simulation as $\rho$ increases from 0 to 0.25. Type 1 error was estimated as the proportion of replicates for which $p \leq 0.05$. 95% confidence intervals were calculated as Wilson score intervals. The dotted line depicts the size of the test, 0.05.
(TIF)

**S1 Text. Supplementary tables A-F.**
(PDF)

**S2 Text. Materials and methods.**
(PDF)

**S3 Text. Supplementary discussion of the declining type 1 error rate of the GPS tests in the context of increasing sample sizes.**
(PDF)

**S1 Data. Supplementary data set containing UKBB case and control overlap, and $\rho$ estimates.**
(TSV)

## Acknowledgments

We would like to thank Dr Xavier Warin for timely assistance with the use of the Stochastic Optimisation library `StOpt` [54]. We would also like to thank our colleague Dr Guillermo Reales for his curation of some of the GWAS data sets used in this work and creation of the `GWAS_tools` pipeline. We wish to acknowledge all GWAS participants, in particular those of the UK Biobank and FinnGen, for their contribution to the data used herein. We also acknowledge the investigators who carried out these GWAS and made their summary statistics publicly available. We acknowledge in particular the Pan-UKBB team [55]. This research has been conducted using the UK Biobank Resource under Application Number 98032.

## Author Contributions

**Conceptualization:** Thomas W. Willis, Chris Wallace.

**Data curation:** Thomas W. Willis.

**Formal analysis:** Thomas W. Willis.

**Funding acquisition:** Chris Wallace.

**Investigation:** Thomas W. Willis, Chris Wallace.

**Methodology:** Thomas W. Willis.

**Software:** Thomas W. Willis.

**Supervision:** Chris Wallace.

**Validation:** Thomas W. Willis.

**Visualization:** Thomas W. Willis.

**Writing – original draft:** Thomas W. Willis, Chris Wallace.

**Writing – review & editing:** Chris Wallace.

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
