## [Decision Letter · Decision Letter 0]

4 Dec 2022

Dear Dr Willis,

Thank you very much for submitting your Methods entitled 'Accurate detection of shared genetic architecture from GWAS summary statistics in the small-sample context' to PLOS Genetics.

The manuscript was fully evaluated at the editorial level and by independent peer reviewers. The reviewers appreciated the attention to an important problem, but raised some substantial concerns about the current manuscript. Based on the reviews, we will not be able to accept this version of the manuscript, but we would be willing to review a much-revised version. We cannot, of course, promise publication at that time.

If you decide to revise the manuscript for further consideration at PLOS Genetics, please aim to resubmit within the next 60 days, unless it will take extra time to address the concerns of the reviewers, in which case we would appreciate an expected resubmission date by email to plosgenetics@plos.org.

We are sorry that we cannot be more positive about your manuscript at this stage. Please do not hesitate to contact us if you have any concerns or questions.

Yours sincerely,

Michael P. Epstein

Academic Editor

PLOS Genetics

David Balding

Section Editor

PLOS Genetics

Reviewer's **Comments to the Authors:**

Reviewer #1: Willis and Wallace propose an improved version of the GPS test. This tests for pleiotrophy between pairs of traits (i.e., whether they have a common genetic architecture). GPS was first proposed (by different authors) in 2015. In this paper, Willis and Wallace explain a fundamental "flaw" in the original version of the test, and also provide a solution. They then compare GPS to alternative methods for detecting shared genetic architecture (e.g., LDSC and Sumher) on both simulated and real data, finding examples where GPS is more powerful.

Overall, I found this a easy paper to read (I thought it was well-written). I considered the modification mathematically sound, and the applications of the method generally extensive. This is not a ground breaking paper, in that the GPS method is not hugely used, and it has limitations (particularly, the sensitivity to individual SNPs, and the lack of an interpretable parameter), all of which are acknowledged by the authors. Nonetheless, the authors do a good job finding situations where it might be useful, and provide extra insight (in particular, for rare traits where there are very few samples). Further, the GPS can always be used (unlike LDSC or Sumher that sometimes fail).

###############

Major comments

I have no major comments. To me the method seemed mathematically sound, and as I say, the authors fairly described its utility and shortcomings.

Minor Comments

1 - Please can you make numbers of SNPs more prominent in the paper. I could not see it specified in the main text. Further, it as also hard to find in the methods (e.g., you described in detail the pruning of 1000G data, but did not mention how many snps remained). For example, perhaps you could add under results a mention that you "usually use LD-pruned data, so n is typically 40k-80k", or whatever the correct numbers are. This would also be useful as computation time presumably is about linear in number of SNPs, so it would make your claim of 1 hour on 8 CPUs more informative.

2 - When describing the simulations, you mention cases numbers, but please can you also mention case control ratio (e.g. do you always have equal cases and controls, or is total sample size fixed). My understanding is that "small-sample" primarily refers to number of cases, rather than small overall sample size. Similarly, you had this sentence "The GPS test identified genetic similarity among IMD pairs across a range of sample sizes" - whereas I felt you were mainly considering different case numbers.

3 - I was confused why the total sample size varied for UKBB traits (especially if you got them from Neale Lab, whom I thought tried to use as many samples as possible). Please can you add a quick comment (sorry if I missed it).

4 - It might be useful to report phenotypic similarity (or average across the four classes, immune/immume, immune/non-immune, more than 2k, less than 2k).

5 - Figure 3 - I assume red line is present in bottom right panels (but cant be seen) - is there some way to make it visible, or perhaps put red line on top (because we care about that method most)

6 - You compare your new version of GPS with other tools. Is it worth also including the original version of GPS (although you have shown it is flawed, it seems it is "conservatively" flawed? - ie tends to give false negatives, rather than false positives, so a comparison would show the improvement in power of your modifications)

7 - There are more details in LDSC and SumHer than most realise. For example, default LDSC estimates genetic correlations across about 8M sequenced snps with MAF>0.005, whereas its default estimates of SNP heritability are across about 4M sequenced SNPs with MAF>0.05 (which in my view is a contradiction, it considers rare SNPs problematic when it comes to SNP heritability, but fine when it comes to GC (or inflation)).

Similarly, for SumHer, estimates depend on choice of heritability model (which you explained), but also on choices of "regression SNPs, heritability SNPs and reference SNPs". Note that I believe you used fewer reference snps for Sumher than LDSC, which might explain its higher power (because fewer reference SNPs leads to slightly smaller SDs)

In theory, you *could* pick these settings to best match your simulated data - ie, work out the assumed heritability model, and the most appropriate regression SNPs, heritability SNPs and reference SNPs. Then your results from LDSC / SumHer would be the best available.

I do not recommend doing this, because in practice, you do not know the truth.

So in summary, I dont think it is necessary / appropriate to go into these technical details (even though they can be important). But instead, I would just acknowledge something along the lines that you consider both LDSC and SumHer because they make contrasting assumptions (in particular, regarding genetic architecture), and thus it is reassuring to see they both perform similarly

Some very minor comments (no responses required)

I thought your description and use of my software Sumher was accurate, many thanks. I also liked the introduction (the mention of hairless dogs, blue-eyed cats was a fun change from the generic introductions that most of us are guilty of writing). On the other hand, I prefer "without" to "sans".

Signed Doug Speed

Reviewer #2: Accurate detection of shared genetic architecture from GWAS summary statistics in the small-sample context

Willis and Wallace

The authors propose a method to access if genetic correlation between two phenotypes is significantly different from zero, by using a modified version of a non-parametric test introduced by Li et al. (2015). The authors argue that an extreme value distribution should be assumed under the null, which is the modification of Li et al. (2015) who used a transformation of the standard exponential distribution. The authors used simulated data as well as real data (GWAS of 18 phenotypes from the UK Biobank) to show that the proposed method outperforms existing genetic correlation methods when using less polygenic traits with a small sample size.

This paper is interesting, but there are a number of questions and comments that should be positively addressed.

1. Eq. (1) is mostly identical with that introduced by Li et al. (2015) (reference #37 in this paper). Eq. (2) is the distribution used in Li et al. (2015). What is the equation used in this study? I am not very clear if the proposed method is substantially improved from the original method or not.

The authors argue that they establish that the null distribution of the test statistic is better modelled by an extreme value distribution than a transformation of the standard exponential distribution. However, they did not really compare the performances for the original and proposed methods. What are type I error rate and power when using a transformation of the standard exponential distribution, and how they are compared with the results shown in Figures 2 and 3 (possibly Table 3)?

2. I am not sure why the type error rates are deviated from the expectation when sample size increases for GPS (Fig. 2). For other methods, it is also unexpected that some data points are significantly deviated from the expectation.

In the legend of Fig. 2, it says, “This proportion was measured only among those replicates producing a valid test statistic.” What does mean by a valid test statistic? How many replicates (%) are invalid and why? It is not clear if this caused the biased type I error (deflation). The authors should make sure the biased type I error rate is not because of this artificial process (excluding some replicates due to some numerical issue that may be correlated with type I error rate).

3. There is no clear detail about Hoeffding’s test. Why they assess the performance of Hoeffding’s test? How are they compared with Li et al.? Does it use a transformation of the standard exponential distribution?

4. Can the authors consider more comprehensive simulations? In Table 1, the numbers of causal variants are small. The authors should consider 1,000 or 10,000 causal variants that may be realistic in many disease traits.

5. The proposed test statistics can used to assess if the genetic correlation is significantly different from 0. Can it be used to test if the genetic correlation is significantly different from 1?

6. Please check references, e.g. there is no journal name in Reference # 33.

Reviewer #3: In order to test for genetic correlation between two traits with two corresponding GWAS summary data, the authors proposed applying an existing bivariate nonparametric method, called GPS, to test the dependence between two sets of the genome-wide p-values for the two traits, as an alternative to the more common approaches of estimating and then testing the genetic correlation directly as implemented in the popular LDSC and SumHer methods/software. A main contribution is to use a GEVD as the null distribution of the test statistic, instead of an exponential distribution in the original GPS paper. Perhaps the most important and interesting message is that, for small sample sizes (i.e. small numbers of cases) with rare diseases, the proposed GPS performed better than LDSC and SumHer in simulations and an application to the UKB GWAS data. The paper was well written, and if the main conclusion of the paper holds, the proposed method will be useful in practice.

Main comments:

1. UKB application: since all the GWAS data/traits are from the UKB, due to overlapping samples, isn’t it expected that the p-values of two traits are possibly non-independent? If so, it would undermine the conclusion of the proposed GPS test. From the methods point, the proposed GPS test requires the two GWAS datasets (or p-values) to be independent, doesn’t it?

2. The null distribution of the GPS statistic: In Fig 1 the authors used the UKB data to show empirically a better approximation of the proposed GEVD over the previously proposed EXP(1); but given the limitations/problems with the use of the UKB data, it may be questionable. Since this is the foundation of the proposed method and a major contribution of the paper, while I suppose that there is no theory to support the use of a GEVD or EXP(1) as the null distribution, some more convincing and extensive evaluations would be helpful.

Relatedly, given that the SNPs are correlated (with the use of the threshold r2=0.2), strictly speaking, even under H0, a permutational distribution is not necessarily the same as the true null distribution because a permutation would destroy the original correlation structures among the p-value pairs; this point can be addressed in simulations to compare the empirical null distribution of the test statistic (when the data are simulated under H0) with the corresponding permutational distribution. It is also somewhat concerning to see that in Fig 2 the type I error of the GPS test goes down so much as the #cases increases, while those for other methods approach the nominal level as expected.

3. How do the test results depend on the LD threshold (r2= 0.2) used in pruning the SNPs? Can the authors do some sensitivity analyses in simulations and real data? This is also related to the reliability of the conclusions with the use of the MHC region in the real data analysis.

**Have all data underlying the figures and results presented in the manuscript been provided?**

Reviewer #1: Yes

Reviewer #2: Yes

Reviewer #3: Yes

PLOS authors have the option to publish the peer review history of their article (what does this mean?). If published, this will include your full peer review and any attached files.

Reviewer #1: **Yes: **Doug Speed

Reviewer #2: No

Reviewer #3: No

---

## [Decision Letter · Decision Letter 1]

17 May 2023

Dear Dr Willis,

Thank you very much for submitting your revised Methods manuscript entitled 'Accurate detection of shared genetic architecture from GWAS summary statistics in the small-sample context' to PLOS Genetics.

If you decide to revise the manuscript for further consideration at PLOS Genetics, please aim to resubmit within the next 60 days, unless it will take extra time to address the concerns of the reviewers, in which case we would appreciate an expected resubmission date by email to plosgenetics@plos.org.

We are sorry that we cannot be more positive about your manuscript at this stage. Please do not hesitate to contact us if you have any concerns or questions.

Yours sincerely,

Michael P. Epstein

Academic Editor

PLOS Genetics

David Balding

Section Editor

PLOS Genetics

Editor comments:

While reviewers 1 and 2 were generally satisfied with the revision, reviewer 3 remains concerned about the impact of sample overlap on the validity of findings and we feel that this topic warrants further response. Specifically we suggest:

Reproduce equation (7) in the Lin and Sullivan AJHG paper that you cite (link here: https://www.ncbi.nlm.nih.gov/pmc/articles/PMC2790578/).Based on this equation and the UKB case-control numbers in Table 2, report the distribution of correlation among effect sizes due to sample overlap across all different combination of traits considered.Perform a simulation exploring the validity of the method under sample overlap at least as great as the "worst" case from 2 above.

The most plausible sample overlap scenario mainly involves controls and we agree this would give little cause for concern. However, shared cases (comorbid phenotypes) can also arise and their effect can be noticeable based on the formula of Lin and Sullivan. This scenario needs to be addressed, briefly but with more detail than currently offered.

Reviewer's **Comments to the Authors:**

Reviewer #1: The authors have comprehensively answered all my comments, thank you. It seems they have also comprehensively answered the comments of the other two reviewers.

Reviewer #2: The authors have addressed most of my concerns. However, based on the new results, there is no clear evidence that the proposed method outperforms existing methods such as SumHer for moderately and highly polygenic traits, as depicted in Fig. 3 with small effect sizes. Additionally, the traits listed in Table 2 related to the immune system appear to be less polygenic than other complex traits (e.g. type 1 diabetes, RA, etc.). Although I believe this work is suitable for publication, it is important that the authors acknowledge this limitation in their Discussion section to prevent potential confusion among readers and users of the proposed method.

Reviewer #3: I appreciate the authors’ responses to my questions/comments. I am fine with most of them except Comment 1, which is about whether/how to account for overlapping samples with the use of the UKB data. Although the authors agree that the overlapping samples would cause non-independence of the sets of the p-values being tested, but stated that

“but this correlation is inversely related to the root of the total sample size of each study 1. Given the large sample sizes of the studies we used, we expect this correlation and its effect on the GPS tests to be negligible.”

I disagree. First, because any two traits were measured from (almost) the same set of the UKB individuals, the overlapping proportion is essentially 1. Hence, no matter how large the sample size, this correlation will NOT disappear. Second, as clearly stated in the cited reference 1 (Lin and Sullivan, AJHG, 2009), “failure to account for overlapping subjects can greatly inflate type I error”. I do not understand why/how the authors claim that “we expect this correlation and its effect on the GPS tests to be negligible”; I’d love to see both theoretical and empirical/simulation results to support their claim. Note that this is a key issue that may implicate the validity or invalidity of the proposed method (regarding to whether it can control the type I error), but has NOT yet been addressed either theoretically or empirically.

**Have all data underlying the figures and results presented in the manuscript been provided?**

Reviewer #1: Yes

Reviewer #2: Yes

Reviewer #3: None

PLOS authors have the option to publish the peer review history of their article (what does this mean?). If published, this will include your full peer review and any attached files.

Reviewer #1: **Yes: **Doug Speed

Reviewer #2: No

Reviewer #3: No

---

## [Editor Report · Decision Letter 2]

30 Jun 2023

Dear Dr Willis,

We are pleased to inform you that your manuscript entitled "Accurate detection of shared genetic architecture from GWAS summary statistics in the small-sample context" has been editorially accepted for publication in PLOS Genetics. Congratulations!

Yours sincerely,

Michael P. Epstein

Academic Editor

PLOS Genetics

David Balding

Section Editor

PLOS Genetics

Comments from the reviewers (if applicable):

**Data Deposition**

http://datadryad.org/submit?journalID=pgenetics&manu=PGENETICS-D-22-01171R2

**Press Queries**

---

## [Editor Report · Acceptance letter]

13 Aug 2023

PGENETICS-D-22-01171R2 

Accurate detection of shared genetic architecture from GWAS summary statistics in the small-sample context 

Dear Dr Willis, 

We are pleased to inform you that your manuscript entitled "Accurate detection of shared genetic architecture from GWAS summary statistics in the small-sample context" has been formally accepted for publication in PLOS Genetics! Your manuscript is now with our production department and you will be notified of the publication date in due course.

With kind regards,

Dorothy Lannert

PLOS Genetics

On behalf of:
